# Skin Manifestations in Patients with Selective Immunoglobulin E Deficiency

**DOI:** 10.3390/jcm11226795

**Published:** 2022-11-17

**Authors:** César Picado, Adriana P. García-Herrera, José Hernández-Rodríguez, Alexandru Vlagea, Mariona Pascal, Joan Bartra, José Manuel Mascaró

**Affiliations:** 1Department of Respiratory Diseases, Hospital Clinic de Barcelona, University of Barcelona, 08036 Barcelona, Spain; 2Institut d’Investigacions Biomèdiques August Pi i Sunyer (IDIBAPS), 08036 Barcelona, Spain; apgarcia@clinic.cat (A.P.G.-H.); jhernan@clinic.cat (J.H.-R.); mpascal@clinic.cat (M.P.); jbartra@clinic.cat (J.B.); 3Department of Pathology, Hospital Clinic of Barcelona, University of Barcelona, 08036 Barcelona, Spain; 4Department of Autoimmune Diseases, Hospital Clinic of Barcelona, University of Barcelona, 08036 Barcelona, Spain; 5Department of Immunology, CDB, Hospital Clinic of Barcelona, University of Barcelona, 08036 Barcelona, Spain; vlajea@clinic.cat; 6Department of Allergy, Hospital Clinic of Barcelona, University of Barcelona, 08036 Barcelona, Spain; 7Department of Dermatology, Hospital Clinic of Barcelona, University of Barcelona, 08036 Barcelona, Spain; mascaro@clinic.cat

**Keywords:** autoimmunity, immunodeficiency, immunoglobulin E, lymphoproliferation, skin

## Abstract

Selective immunoglobulin E deficiency (SIgED) is still an unrecognised primary immunodeficiency despite several observations supporting its existence. This study aimed to describe the skin manifestations associated with SIgED. We retrospectively assessed medical records of patients with SIgED, the diagnosis being based on serum IgE levels ≤2 Uk/L associated with normal serum levels of immunoglobulins G, M, and A. A total of 25 patients (24 female) with SIgED were included in the study. Eleven patients (44%) presented chronic spontaneous urticaria (CSU), five (20%) angioedema always associated with CSU, five erythema (20%), and six eczema (24%). Other, less frequent manifestations were lichen planus, anaphylactoid purpura, thrombocytopenic purpura, bullous pemphigoid, bullous pyoderma gangrenosum, and atypical skin lymphoproliferative infiltrate associated with reactive lymphadenopathy, chronic cholestasis, arthritis, and fibrosing mediastinitis. Fifteen patients (60%) had different types of associated autoimmune diseases, Hashimoto’s thyroiditis being the most frequent (*n* = 5, 20%), followed by arthritis (*n* = 4, 16%), autoimmune hepatitis, neutropenia, vitiligo, and Sjögren’s syndrome (*n* = 2, 8% each). Five malignancies were diagnosed in four patients (16%). An ultralow IgE serum level may be the only biomarker that reveals the presence of a dysregulated immune system in patients with a broad spectrum of skin manifestations.

## 1. Introduction

Inborn errors of immunity result from mutations in one or various genes required for normal function of the immune system [1,2]. To date, more than 400 distinct immunodeficiency diseases have been identified, including disorders associated with antibody deficiencies [1,2]. Common variable immunodeficiency disorder (CVID), selective immunoglobulin A deficiency (SIgAD), selective immunoglobulin M deficiency (SIgMD), and subclasses of immunoglobulin G deficiency are among the immune diseases included in the antibody deficiency group [1,2]. 

Although low serum levels of immunoglobulin E (IgE) are associated with CVID in some patients [3,4], the existence of an isolated and selective IgE deficiency (SIgED) has not yet been recognised as a distinct entity in the recently updated classification of inborn errors of immunity [1,2]. However, various studies [5,6,7,8,9] have reported clinical observations supporting the notion that SIgED has a potential role in the immune system similar to other accepted selective immunoglobulin deficiencies such as SIgMD and SIgAD [10,11,12]. Interestingly, various recent epidemiological studies have found an inverse relationship between serum IgE levels and malignancy development [13], an association previously described in primary immunodeficiency diseases (PIDs) [14]. 

Skin manifestations are common in PIDs [15]; however, most skin manifestations found in these processes can also be seen in patients with an apparently normal immune system. 

Predominantly cutaneous manifestations in the antibody deficiency group include skin infections, dermatitis-like lesions, chronic spontaneous urticaria (CSU), eczema, psoriasis, atopic dermatitis, seborrheic dermatitis, alopecia, vitiligo, and oral ulcers [15,16,17,18]. 

The objective of the present study was to describe the clinical and histopathological characteristics of the skin manifestations found in patients with SIgED.

## 2. Patients and Methods

A total of 25 adult patients (24 females), mean age 68.5 years (range 29–90), diagnosed with SIgED at the Hospital Clinic (University of Barcelona, Barcelona, Spain), were enrolled in the study. Clinical and laboratory data were retrospectively collected and the diagnosis revised by their treating physicians. SIgED deficiency was established based on the presence of an IgE serum level of ≤ 2kU/L associated with normal serum levels of IgA, IgM, and IgG. Patients with potential secondary causes of hypogammaglobulinemia (e.g., rituximab treatment) were excluded. None of patients were on treatment with drugs such as dupilumab that can potentially reduce IgE serum levels. Some of these patients had been previously reported in a recent publication [9]; however, neither the dermatologic nor the dermatopathological features of the skin manifestations were described in detail. 

## 3. Results

Skin manifestations are shown in Table 1, the most frequent being CSU (*n* = 11, 44%), angioedema (AD) always associated with CSU (*n* = 5, 20%), isolated erythema or associated with other manifestations (*n* = 5, 20%), eczema (*n* = 6, 24%), and purpura (*n* = 2, 8%). In one of the two patients with purpura (No. 20), the skin manifestation was preceded by a rash with a hive-like appearance that rapidly changed to palpable purpura, associated with arthritis affecting the ankles, abdominal pain, and enterorrhagia. This association of diverse clinical manifestations is very typical of anaphylactoid purpura. In the second patient with purpura (No. 21), this resulted from immune-induced thrombocytopenia. Other less frequent skin manifestations are depicted in Table 1. Fifteen patients (60%) had different types of autoimmune diseases, Hashimoto’s thyroiditis being the most frequent (*n* = 5, 20%), followed by arthritis (*n* = 4, 16%), autoimmune hepatitis, neutropenia, vitiligo, and Sjögren’s syndrome (*n* = 2, 8% each) (Table 1). Five malignancies were diagnosed in four patients (16%), the most common malignancy being breast cancer (*n* = 3, 12%). Other associated diseases are shown in Table 1. 

A skin biopsy was performed in six patients. In two (Nos. 5 and 8), the biopsy revealed mild lymphocytic perivascular inflammatory infiltrate in superficial and middle dermis associated with eosinophils and few neutrophils. Direct immunofluorescence (DIF) examination was negative, and a clinicopathological diagnosis of urticaria was carried out.

Patient No. 4 was a 43-year-old woman with autoimmune hepatitis and an eight-year history of episodes of non-itching hives lasting 24 h or more, with residual hyperpigmentation associated with joint pain. The skin biopsy demonstrated a moderate perivascular inflammatory infiltrate made up of lymphocytes and eosinophils in the middle and deep dermis. Focal changes in erythrocyte extravasation and karyorrhexis were also seen in the deep dermis. A clinicopathological diagnosis of urticarial-vasculitis was decided.

Patient No. 12 was a 70-year-old woman with a 10-year history of itching non-specific dermatitis associated with chronic scratching on both lower extremities with poor response to antihistamine and local glucocorticoid therapy. The skin biopsy revealed epidermal acanthossis with orthokeratotic hyperkeratosis, mild inflammatory lymphocytic infiltrates in the upper dermis with vascular hyperplasia, and mild fibrosis. A clinicopathological diagnosis of lichen simplex chronicus was made, and the patient was successfully treated with methotrexate. 

Patient No. 22 was an 82-year-old woman presenting with bullae arising on an erythematous base on her lower right extremity. She had experienced a similar episode 10 years earlier, which healed completely after treatment with topical corticosteroids for several weeks. A cutaneous biopsy showed epidermal spongiosis, dermal oedema, and superficial perivascular lymphocytic infiltrates with abundant eosinophils. DIF examination of perilesional skin revealed linear basement membrane zone deposits of IgG and complement component 3 (C3). A diagnosis of bullous pemphigoid was made, and her cutaneous lesions healed completely after treatment with topical corticosteroids. 

Patient No. 23 was a 90-year-old man. His first skin lesions developed five years before in the form of erythematous plaques with some pustules that later progressed to a chronic painful ulcer on the lower leg. A biopsy obtained from the edge of the ulcer revealed a partially ruptured subepidermal blister, with dense neutrophilic infiltrates in the superficial dermis, extending towards the middle dermis. A clinicopathological diagnosis of bullous pyoderma gangrenosum was made. Slight improvement was observed after topical treatment combined with oral prednisone and cyclosporine. Subsequently, a *pseudomonas aeruginosa* infection of the ulcer developed that required antibiotic treatment. Due to the side effects of cyclosporin, treatment with mycophenolate mofetil was initiated; despite this, the ulcer never healed. She developed several complications (diabetes, peripheral arterial ischemia, and heart failure) that contributed to her death. 

Patient No. 24 was a 46-year-old woman who started smoking at the age of 13 years with irregular, but recurrent, episodes of fever, right-side abdominal pain, and polyarthritis mainly affecting the small joints of the hands and knees, which improved with glucocorticoid treatment and relapsed when treatment was discontinued. Abnormal liver function tests were found, with high levels of alkaline phosphatase (ALP, 755 U/L, normal upper limit 116 U/L), alanine transaminase (161 U/L, normal upper limit 40 U/L), and aspartate transaminase (124 U/L, normal upper limit 40 U/L). The patient was treated with ursodeoxycholic acid (UDCA), and a cholecystectomy was finally performed. After that, the episodes of fever and abdominal pain appeared to be less frequent (every three to six months) but required regular treatment with glucocorticoids. At the age of 23 years, viral markers for hepatitis were all negative and screening for systemic and autoimmune liver disease was also negative. An abdominal ultrasound showed gallstones and splenomegaly. Several examinations were made throughout the follow-up, including mutation in the *ABCB4* gene, computed tomography, magnetic resonance cholangiopancreatography, ^18^F-fluorodeoxyglucose–positron emission tomography (FDG-PET), and liver biopsy. Imaging studies only demonstrated a slight segmental dilation of the intrahepatic bile duct and the presence of numerous abdominal and retroperitoneal lymphadenopathies that showed hypermetabolic activity in the FDG-PET study. The liver biopsy only revealed a slight increase in the number of bile ducts in some portal spaces. Abdominal lymph node excision was performed, and the histological study showed reactive lymphadenopathy, negative for malignancy. 

At the age of 40 years, the genetic study of monogenic autoinflammatory diseases did not find pathogenic variants in *MEFV*, *TNFRSF1A*, and *NLRP3* genes. An heterozygous variant p.(Asn289Ser) in the *NOD2* gene was detected. However, this non-pathogenic variant confers susceptibility to develop inflammatory bowel disease. Because the phenotype expressed by the patient (recurrent fever and other inflammatory manifestations) did not fit with any of the monogenic autoinflammatory conditions studied, she was considered to have an undifferentiated autoinflammatory disease. Colchicine (1 mg/day) was then started with good control of febrile episodes and joint pain.

At the age of 42 years, she developed an erythematous non-itchy, hardened, and painful skin lesion 7 mm in length on the postero-lateral area of her right thigh with extension to deep tissues and no systemic symptoms. The inflammatory lesion improved with prednisone at 30 mg/day, but local relapses occurred when the dose was tapered below 15 mg. A large biopsy including skin, quadriceps, and fascia showed multiple foci of perivascular lymphoid inflammatory infiltrate affecting the dermis spreading into subcutaneous cellular tissue and infiltrating fascia and muscle. No destruction of anatomical structures, nor epidermotropism or folliculotropism was observed. Small-vessel vasculitis in the dermis and adipose tissue was associated with thrombosis and signs of recanalisation in isolated vessels. Moderate increases in spindle stromal cells with large nuclei, open chromatin, and isolated figures of mitosis were also present. The cellular infiltrate was composed of small lymphocytes (without atypia) and focal nodule formation but no germinal centres, and few plasma cells, eosinophils, and neutrophils. An immunohistochemical study detected lymphoid infiltrates with predominant T cells without phenotypic abnormalities (CD3-, CD2-, CD5-, and CD7-positive), with a predominance of CD8/CD4 and co-expression of Bf1 (delta-chain staining negative). The small nodules corresponded to CD20 B lymphocytes. Plasma cells within the infiltrates were polytypic, and only a few isolated cells expressed IgG4. Stromal cells were CD34+, and smooth muscle actin was negative. The final histopathological diagnosis was that of an atypical lymphoid infiltrate, without phenotypic abnormalities or clonality, associated with small-vessel vasculitis involving dermis, fascia, and muscle.

Local low irradiation therapy (20Gy) was indicated with little effect, since it was necessary to continue regular treatment with prednisone at daily doses of 10 mg or higher. Six weeks later, the irradiated area began to show signs of inflammation and fluctuation, which led to an ulcer 3 cm in diameter. After two and a half years and several relapses following decreases in prednisone dose to below 10 mg/day, subcutaneous methotrexate at 12.5 mg/week (0.3 mg/kg) was started with good clinical control. The patient was then able to achieve continuous prednisone tapering to 2.5 mg/day. Colchicine was subsequently stopped at the age of 44 years.

At the age of 45 years, she presented with persistent cough and right thoracic pain over three weeks. A computed tomography scan revealed a diffuse infiltrative mediastinal lesion extending towards the paravertebral region and the right posterior and lateral part of the trachea. It also spread in the form of a cuff through the right main bronchus and other bronchi (upper lobar and intermediate). A wide extension of the right pleural space was also observed. Numerous mediastinal lymph nodes with a tendency to converge with the infiltrative process were noted. FDG-PET imaging only showed moderate hypermetabolic activity in the mediastinal lymph nodes. Endobronchial-ultrasound-guided fine-needle aspiration of mediastinal lymph nodes only revealed reactive lymphadenopathy, negative for malignancy. A CT-guided percutaneous needle biopsy for the mediastinal infiltrative lesion revealed connective tissue with fibrous changes and the presence of a mixed infiltrate composed of mature lymphocytes, histiocytes, and plasma cells. No granulomas were seen. Immunohistochemistry showed inflammatory infiltrates with predominantly T lymphocytes and, to a lesser extent, B lymphocytes. Plasma cells were negative for IgG4. The histopathological diagnosis was “inflammatory pattern of fibrosing mediastinitis and no malignancy”. The patient is currently being treated with methotrexate 10 mg/week and prednisone with no thoracic or systemic complaints, and without apparent progression in the size of the mediastinal process. She has maintained treatment with UDCA and a progressive decrease in alkaline phosphatase values has been noted throughout the process, without reaching normal levels (last determination 188 U/L). No flares of fever or musculoskeletal manifestations have occurred since colchicine cessation. 

## 4. Discussion

As with other antibody deficiencies and SIgED, the number of women was much higher than that of men in the patients included in this study [3,4,6,8,9,10,12]. In our cohort, CSU, associated or not with AD, was a frequent skin manifestation in patients with SIgED. In keeping with our findings, a higher prevalence of CSU has been previously reported in patients with SIgED (19%) compared with controls (0.8%) [8]. 

CSU and AD are not usually found among the most common clinical manifestations in CVID [19]; however, there are reports suggesting that the association might be underdiagnosed [20,21,22,23]. The prevalence of CSU has also been found to be higher in patients with SIgAD (4.9%) than in controls (0.9%) [18], and in up to 12% of patients with SIgMD [16].

Non-specific eczema and erythema, usually associated with itching, are also highly prevalent skin manifestations among our SIgED patients. Severe eczema and erythema are characteristic of numerous diseases associated with immune dysregulation, particularly those diagnosed in early life [15,24,25]. In contrast, non-specific eczema and erythema do not usually appear among the skin symptoms most frequently found in adults suffering from CVID, SIgMD, SIgAD, and other diseases associated with immune dysregulation. The reasons accounting for this difference observed between children and adults are unclear. Interestingly, severe eczema is a very frequent cutaneous manifestation in patients with either autosomal dominant or autosomal recessive forms of hyper-IgE syndrome [26]. The mechanisms by which both ultralow and ultrahigh levels of IgE are associated with eczema and erythema, often severe, are currently unknown. 

Two of our patients suffered from bullous pemphigoid and bullous pyoderma gangrenosum, two skin diseases that have been described in association with CVID [27,28,29,30] and SIgAD [31] in a few isolated cases. Pyoderma gangrenosum has been previously reported in a patient with SIgED [32].

Purpura was the skin manifestation found in two patients, in one case being associated with symptoms considered pathognomonic of anaphylactoid purpura, a disease not previously reported to be associated with immunodeficiency disorders. In contrast, in the second patient, purpura was associated with an immune thrombocytopenia, which is frequently found in patients with CVID [19], but only occasionally reported in patients with other selective antibody deficiencies [10,33]. 

Patient No. 24 is a complex case that could not be assigned to any of the clinical entities included in the classification of inborn errors of immunity [1,2], since she only presented a selective deficiency of IgE, which, as previously mentioned, is an immunodeficiency that is not recognised in the aforementioned classification. 

In her teens, she underwent health examinations for complaints suggestive of a disease of the bile ducts associated with liver disease, periodic fever, and polyarthritis. However, the exhaustive analyses and explorations carried out did not allow a diagnosis to be established beyond verifying the existence of idiopathic cholestasis. Regular treatment with glucocorticoids and UDCA was required to achieve relative control of her symptoms. In addition, the image studies revealed the presence of splenomegaly and abdominal and retroperitoneal lymphadenopathies; a biopsy of one of them showed benign lymphoproliferation. Thirty years later, she developed an infiltrative skin lesion with involvement of the subjacent quadriceps muscle and fascia, the biopsy of which showed an inflammatory lymphoproliferative process of all layers with small-vessel vasculitis. The latest manifestation of the disease was a marked enlargement of the mediastinal lymph nodes, a pathological examination of which revealed a reactive lymphoid proliferation without malignant features. A mediastinal biopsy concluded with the diagnosis of fibrosing mediastinitis associated with the enlarged lymph nodes. 

If the study of this patient had shown low serum levels of IgG and IgM and/or IgA, we would have concluded that, of the whole hepatobiliary process, undifferentiated arthritis, splenomegaly, and the lymphoproliferative processes of the skin, muscle, fascia, lymph nodes, and mediastinum were most probably due to CVID. 

Increased liver enzyme levels, including ALP, are frequent in CVID. Some CVID patients with abnormal liver enzymes suffer from autoimmune hepatitis, primary sclerosing cholangitis, and even liver cirrhosis [34,35]; however, in some patients, abnormal liver enzymes cannot be linked to any specific disease. Lymphoid hyperplasia (30%) and splenomegaly (29%) are found in almost a third of CVID patients [19], and rheumatoid arthritis and other non-specific arthritides are also diagnosed in 3% of these patients [19]. 

Although benign, the cutaneous lymphoproliferative process and lymphadenopathy of this patient deserve a polarised discussion. Lymphoproliferative disorders (LPDs) in immunodeficient patients range from benign to malignant lymphoid proliferation. LPD associated with primary immune disorders (PIDs) is one of the four types of LPDs recognised in the World Health Organization classification [36]. Biopsies of lymph nodes from LPDs in patients with PIDs usually show reactive lymphoid hyperplasia, atypical lymphoid hyperplasia, or granulomatous inflammation [37,38]. The presence of lymphadenopathies in a patient with an immunodeficiency poses a significant diagnostic dilemma, the differentiation between benign proliferation and lymphoma, because the finding of lymphadenopathy in a patient with CVID indicates an increased susceptibility to the development of both nodal and extra-nodal lymphomas, particularly non-Hodgkin’s lymphomas [37,38,39,40]. However, careful distinction between malignant and non-malignant lymphoid proliferation in CVID is not simple, as clonal expansions can occur in non-malignant nodes, tissues, and blood in patients with CVID [36,37,38,39,40]. 

Extra-nodal lymphoproliferative disorders are frequent in CVID patients, the lung and gastrointestinal tract being more frequently affected than skin [37,38,39,40,41]. When lymphoid proliferation arises in the skin of patients with CVID, it poses the same difficulties in differentiating benign from malignant lymphoproliferative disorders, as occurs when the same process affects the lymph nodes [42].

The lymphoproliferative disorders affecting the lymph nodes and the skin of patient No. 24 are quite similar to those reported in CVID patients. What is relevant in our observation is that the serum levels of IgG, IgM, and IgA were normal during many years of follow-up, and, similarly, the ultralow IgE levels remained unchanged throughout the follow-up.

Interestingly, the fibrosing mediastinitis recently diagnosed in the patient has also been described preceding a subsequent diagnosis of lymphoma, a complication that has not been detected in our patient during the follow-up [43,44,45].

This patient presented clinical manifestations suggestive of an autoinflammatory disease. Interestingly, lymphadenopathy and benign lymphoproliferation can be part of the spectrum of both PIDs and autoinflammatory processes [46]. Whether SIgED is associated with autoinflammatory disorders is unknown and needs to be examined in further studies. 

Our study has several limitations, including its retrospective nature, the collection of experiences from a single centre, and that a selection bias for SIgED patients cannot be excluded. In addition, the number of biopsies is low, largely because in some common skin manifestations, such as CSU, eczema/dermatitis, psoriasis, vitiligo, or erythema, the diagnosis is usually made clinically, and biopsies are not routinely used for evaluation.

The high prevalence of autoimmune diseases and malignancies found in our patients with SIgED has been previously reported [6,8,9,13].

In summary, SIgED is associated with numerous skin diseases, most of which have been described as part of the manifestations of various immunodeficiency disorders such as CVID, SIgAD, and SIgMD. Although SIgED is not yet recognised as an immunodeficiency with its own distinctive profile, evidence to date suggests that IgE testing should be included in the work-up of patients whose disease is suspected to be related to immune system dysregulation. 

## Figures and Tables

**Table 1 jcm-11-06795-t001:** Distribution of skin manifestations, autoimmune diseases, and malignancies in patients with SIgED.

No.	Age/Sex	Skin Manifestations	Diagnosis	Autoimmunity	Other Diseases	Tumours
1	64/F	Hives/edema	CSU/AD		MGUS	
2	65/F	Hives/edema	CSU/AD	Neutropenia Hashimoto’s disease		
3	67/F	Hives	CSU	Enteropathy		
4	43/F	Hives/edema	Urticarial vasculatis	Autoimmune Hepatitis		
5	60/F	Hives/edema	CSU/AD	Hashimoto’s disease Vitiligo	Rhinitis	
6	66/F	Hives	CSU		Asthma	Breast
7	49/F	Hives. Exanthema	CSU Exanthema	Sjögren’s syndrome		
8	60/F	Hives/edema	CSU/AD			
9	68/F	Hives, Erythema	CSU. Erythema	Undifferentiated arthritis	Rhinitis	Basal cell carcinoma
10	51/F	Hives	CSU	Autoimmune hepatitis		
11	43/F	Eczema	Eczematous dermatitis		Thrombophilia. Raynaud’s syndrome	
12	70/F	Eczema	Lichen simplex chronicus	Hashimoto’s disease	Thrombophilia. Bronchiectasis.	
13	85/F	Erythema	Erythema		MGUS	
14	88/F	Eczema. Erythema	Eczematous dermatitis. Erythema	Hashimoto’s disease		
15	79/F	Exanthema	Exanthema		Rhinitis. Otitis	
16	60/F	Erythema.	Erythema	Sjögren’s syndrome. Undifferentiated arthritis. SLE	Lymphocytic interstitial lung disease	
17	63/F	Eczema	Eczematous dermatitis	SLE	Bronchiectasis	
18	29/F	Eczema	Eczematous dermatitis	Neutropenia. Enteropathy	Asthma Rhinitis	
19	79/F	Exanthema	Exanthema		Rhinitis. Otitis	
20	36/F	Purpura	Anaphylatoid purpura	Graves’ disease	Asthma	
21	73/F	Purpura	Thrombocytopenic purpura			Breast. Endometrial carcinoma
22	82/F	Erythema. Bullae	Bullous pemphigoid	Rheumatoid arthritis	Interstitial lung fibrosis	Breast
23	90/M	Erythema. Ulcer	Bullous pyoderma gagrenosum	Hashimoto’s disease	MGUS	
24	45/F	Hardened painful erythematous plaques. Ulcer	Atypical lymphoid infiltrate	Undifferentiated arthritis. Vitiligo.	Chronic cholostasis	Fibrosing mediastinatis

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
