# Peer review of "Skin Manifestations in Patients with Selective Immunoglobulin E Deficiency"

_jcm, 2022, doi:10.3390/jcm11226795_

Round 1

Reviewer 1 Report

This manuscript addresses skin, as well as other clinical manifestations, in a collection of patients with very low IgE levels, deemed selective IGE deficiency (SIgED). The first issue is whether SIgED is an entity, and it is currently not recognized in the inborn errors of immunity listings. There are recent descriptions with infections, but the driver of this is not clear- why low IgE would lead to infections, or if this is a marker of something else, so the comparisons to other antibody deficiencies were a bit tough for me to accept.  CVID has set criteria for inclusion, and I do not know that this does, and so makes it harder to then group the patients together without further details.  For instance, were they on any immune regulatory medications (such as steroids) that may affect serum IgE levels? Were other immune parameters normal including B cells numbers? Was whole exome sequencing performed- just with the autoimmunity as well as other findings, would be good to genetically rule out other IEIs…as well as cancer predisposition genes for the patient with breast and endometrial cancer.  Two patients had MGUS- did this come second to the low IgE or was this present throughout as far as known and could the MGUS cause a secondary IgE? 

I also think in the discussion it is difficult to compare so much to CVID or SIgAD as this entity sounds very different and is not considered a PID. 

More specific comments- I think the description of case 24 is too long. 
Finally, the manuscript is about skin manifestations- how about photos of the rashes? 

Reviewer 2 Report

General comments

This paper summarizes the clinical and histopathological characteristics of the skin manifestations found in patients with selective immunoglobulin E deficiency (SIgED).  SIgED is very rare disease and this paper is valuable. However, some corrections are needed, and the authors should also correct the points listed below.

Specific comments

1, Why does this disease occur predominantly in females? The reason for this has not been discussed.

2, The authors describe that the mechanisms by which both ultralow and ultrahigh levels of IgE are associated with eczema and erythema, are currently unknown.  However, although the cause is unknown, there must be an assumed mechanism for the appearance of eczema and erythema due to the presence of IgE.  Please describe more about the possible pathogenic mechanisms.

Round 2

Reviewer 1 Report

The manuscript is improved.  Although the authors make good points about the existence of a hypo IgE syndrome, there remains a lot to be done in terms of the definition and so case series are a bit tricky.  I would just clear up in the methods the immune suppression that was excluded- potentially state if systemic steroids were allowed during the finding of low IgE and exclusion of other medications that could lower IgE such as dupilumab. 

Author Response

Please see the attchment
